# Overexpression of *GmCAMTA12* Enhanced Drought Tolerance in Arabidopsis and Soybean

**DOI:** 10.3390/ijms20194849

**Published:** 2019-09-29

**Authors:** Muhammad Noman, Aysha Jameel, Wei-Dong Qiang, Naveed Ahmad, Wei-Can Liu, Fa-Wei Wang, Hai-Yan Li

**Affiliations:** College of Life Sciences, Engineering Research Center of the Chinese Ministry of Education for Bioreactor and Pharmaceutical Development, Jilin Agricultural University, Changchun 130118, Jilin, China

**Keywords:** arabidopsis, CaM (Calmodulin), calmodulin-binding transcription activators (CAMTA), *cis*-elements, drought, qPCR, soybean hairy roots

## Abstract

Fifteen transcription factors in the CAMTA (calmodulin binding transcription activator) family of soybean were reported to differentially regulate in multiple stresses; however, their functional analyses had not yet been attempted. To characterize their role in stresses, we first comprehensively analyzed the *GmCAMTA* family in silico and thereafter determined their expression pattern under drought. The bioinformatics analysis revealed multiple stress-related *cis*-regulatory elements including *ABRE*, *SARE*, *G-box* and *W-box*, 10 unique miRNA (microRNA) targets in *GmCAMTA* transcripts and 48 proteins in GmCAMTAs’ interaction network. We then cloned the 2769 bp CDS (coding sequence) of *GmCAMTA12* in an expression vector and overexpressed in soybean and Arabidopsis through *Agrobacterium*-mediated transformation. The T3 (Transgenic generation 3) stably transformed homozygous lines of Arabidopsis exhibited enhanced tolerance to drought in soil as well as on MS (Murashige and Skoog) media containing mannitol. In their drought assay, the average survival rate of transgenic Arabidopsis lines OE5 and OE12 (Overexpression Line 5 and Line 12) was 83.66% and 87.87%, respectively, which was ~30% higher than that of wild type. In addition, the germination and root length assays as well as physiological indexes such as proline and malondialdehyde contents, catalase activity and leakage of electrolytes affirmed the better performance of OE lines. Similarly, *GmCAMTA12* overexpression in soybean promoted drought-efficient hairy roots in OE chimeric plants as compare to that of VC (Vector control). In parallel, the improved growth performance of OE in Hoagland-PEG (polyethylene glycol) and on MS-mannitol was revealed by their phenotypic, physiological and molecular measures. Furthermore, with the overexpression of *GmCAMTA12*, the downstream genes including *AtAnnexin5*, *AtCaMHSP*, *At2G433110* and *AtWRKY14* were upregulated in Arabidopsis. Likewise, in soybean hairy roots, *GmELO*, *GmNAB* and *GmPLA1-IId* were significantly upregulated as a result of *GmCAMTA12* overexpression and majority of these upregulated genes in both plants possess CAMTA binding *CGCG/CGTG* motif in their promoters. Taken together, we report that *GmCAMTA12* plays substantial role in tolerance of soybean against drought stress and could prove to be a novel candidate for engineering soybean and other plants against drought stress. Some research gaps were also identified for future studies to extend our comprehension of *Ca-CaM-CAMTA*-mediated stress regulatory mechanisms.

## 1. Introduction

A successful sustainable agriculture should ensure food security, must be eco-friendly and safe to humans [1]. With the existing population growth rate, the current food production rate needs to be increased at least up to 70% by 2050 [2,3]. Despite advanced farming practices, the abiotic stresses due to drought, salinity, water and temperature fluctuations are causing 50–80% losses in crop yield, and therefore, should be as effectively managed as possible [4]. Soon, the warmer earth will cause a more humid atmosphere but less humid soil, leading to more frequent drought that would negatively affect the rate of photosynthesis, uptake of CO_2_, accumulation of biomass and yield [5,6]. Thus, developing stress-resistant crops with stable yields under adverse conditions is an important strategy to ensure future food security [2,7]. Deep insights into the mechanisms underlying signaling crosstalk mediating stress tolerance would aid plant researchers to upgrade the plant’s indigenous natural machinery through biotechnology and genetic engineering.

In plants, the ubiquitous secondary messenger calcium is the key to maintain the harmonious and homeostatic conditions via signaling [8,9]. Calcium ions perceive and encode the environmental, developmental or hormonal signals into a definite frequency that is decoded and relayed by the protein molecules next to them including calmodulin (CaM). By interacting with calcium, calmodulin (calcium-modulated) proteins sense and convey the signals to calmodulin-binding proteins [10]. Here, CaM through signal frequency readjustment channelizes them before transducing to CAMTA TFs (transcription factors), thus CaM acts like a prism (as prism dissects white light into its components). A transcription factor family reported and initially referred to as Ethylene-induced calmodulin-binding protein [11] or signal responsive (SR) protein, while now known as Calmodulin-binding transcription activator (CAMTA), is present in almost all eukaryotes. CAMTAs were first detected in *Nicotiana tabaccum* while studying calmodulin-binding proteins [11,12,13]. After their emergence, all multicellular eukaryotes studied to date have been reported to be equipped with variable number of *CAMTA* genes such as *Arabidopsis thaliana* (6) [13], *Lycopersicum esculantum* (7) [14], *Medicago truncatula* (7) [15], Citrus (9) [16], *Populus trichocarpa* (7) [17], *Nicotiana tabacum* (13) [18], *Musa acuminata* (5) [19] and *Phaseolus vulgaris* (8) [20]. Glycine max possesses 15 *CAMTA* genes and all of them differentially express under various stress conditions [21].

CAMTA TFs are an integral element of Calcium-mediated biotic/abiotic stress and hormonal signaling pathway [22,23,24,25]. Stress signals are conveyed and modulated through *Ca-CaM-CAMTA* pathway and a rapid and calculated response is observed by carrying out the transcription of stimulus-specific genes [26]. Calmodulin Binding Transcription Activators through their CG-1 domain recognize and specifically bind the ((*A/C*)*CGCG*(*C/G/T*), (*A/C*)*CGTGT*)) sequence of the target genes thereby directly interacting and regulating their transcription [12]. The CG-1 motif is a pivotal member of a rapid stress response element (RSRE) found in the promoters of many genes that are rapidly activated in response to stress [27]. Previous experiments report the negative role of Arabidopsis *CAMTA3* in regulating plant immunity as demonstrated in the *AtMCAMTA3* (loss-of-function) mutant Arabidopsis [28,29,30]. Similarly, *AtCAMTA1* and *AtCAMTA3* have been shown to function in drought and regulate auxin [22,31]. Moreover, *AtCAMTA1* and *AtCAMTA3* regulated cold response by inducing CBF (C-Repeat/DRE-Binding Factor) pathway genes as the double *ATCAMTA1* and *ATCAMTA2* mutant exhibited impaired freezing tolerance [32,33]. The role of *AtCAMTA3* in regulating SA (salicylic acid) pathway genes working in freezing tolerance was only recently determined [34]. Two recent studies highlighted the role of *AtCAMTA3* [35] and *AtCAMTA6* [36] in salt tolerance. *TaCAMTA4* in wheat was demonstrated to negatively regulate defense response against *Puccinia triticina* [25].

Soybean is an important crop cultivated globally for food, feed [37], pharmaceutical and soil nitrogen improving purposes [38]. However, environmental adversities including drought affect soybean growth and yield. Earlier, the in silico analysis of *GmCAMTAs* was conducted, yet the potential targets of miRNA in *GmCAMTA* transcripts and the protein-protein interaction network were not reported [21]. In addition, the previous study also did not analyze the expression pattern of *GmCAMTAs* in soybean leaves in response to drought [21]. In an extension to that study and in order to decipher the role of soybean CAMTA family in drought, we first comprehensively analyzed (in silico) the *GmCAMTA* family including their physicochemical properties, chromosomal distribution, *cis*-motifs, miRNA targets and protein-protein interaction network. Secondly, we determined the spatiotemporal expression pattern of GmCAMTAs in roots and leaves of soybean under PEG stress and selected an efficient member of the GmCAMTA family to functionally characterize. We constructed the overexpression construct by cloning the 2769 bp CDS (coding sequence) of *GmCAMTA12* and transformed into Arabidopsis and soybean hairy roots. Through various drought assays, we demonstrated that the transgenic Arabidopsis and chimeric soybean (OE-Overexpressing *GmCAMTA12*) plants exhibited enhanced tolerance and performed better under drought stress than their non-transgenic counterpart at phenotypic, physiological and molecular level. qPCR (quantitative PCR) of the downstream genes in Arabidopsis and soybean also displayed altered expression as a result of *GmCAMTA12* overexpression. From our analyses, we report that *GmCAMTA12* as a transcription factor plays role in drought stress by regulating the downstream genes involved in drought tolerance and could be exploited in developing drought-tolerant crops.

## 2. Results

### 2.1. Physico-Chemical Properties of GmCAMTA Proteins

Various physico-chemical properties including the number of amino acids, protein molecular weight (MW), pI (isoelectric point), number of atoms, instability and aliphatic indexes and GRAVY (grand average of hydropathy) determined online with the ProtParam tool are given in Appendix A. GmCAMTA11 and GmCAMTA9 are the shortest polypeptides comprising of 910 and 911 aa (amino acid), while GmCAMTA1 is the longest possessing 1122 aa. Overall, their average length is ~1004 aa with a range of ~200 aa mutual difference. The molecular weight of GmCAMTAs ranges between 126,989.43 and 102,394.56 kDa with an average MW of 112848.3 and the number of atoms is proportional to the molecular weight of each protein. Similarly, GmCAMTA8 has the highest pI value of 7.64 showing that GmCAMTAs have relatively lower pI. Moreover, they are also hydrophilic in nature as the GRAVY ranges between –0.625 (GmCAMTA1) and –0.394 (GmCAMTA12). Almost all of the GmCAMTAs are thermally stable as their aliphatic indexes match with that of the other globular proteins (highest in GmCAMTA12). While none of them are stable in a test tube, as the instability index of all GmCAMTAs is higher than 40. Except GmCAMTA8, the number of Asp+Glu (negatively charged aa) is higher than Arg+Lys (positively charged aa).

### 2.2. Phylogenetics and Structure of GmCAMTAs

As mentioned earlier, CAMTAs are multiple-stress responsive transcription factors. Enrichment of *cis*-motifs involved in signal response in the promoters of Medicago *CAMTA* genes hints that they are likely to respond variedly to various signals like other *CAMTAs* [15]. We dissected the regulatory region of *GmCAMTAs* (~2 kb upstream) online with PLANTCare, which detected stimulus-specific *cis*-motifs in their promoters. Overall, there are light (*G-box, MRE* and *AE-box*), drought (*MBS*), salt *(MYB*), pathogen (*TC-rich repeats*), wound (*WUN, WRE*), low temperature (*LTR*), gibberellin (*GARE, P-box*), auxin (*AUXRE*) and abscisic acid (*ABRE*) responsive *cis*-elements as shown in figure 1A. The presence of multiple *cis*-motifs in *GmCAMTA* genes represents their responsivity to multiple stimuli. Moreover, the *cis*-element (*CG-box*) is the binding site of CAMTA TFs; thus, the presence of *G-box* within *GmCAMTA* promoters indicates the interaction of one GmCAMTA TF with another. *GmCAMTA12* possesses *MYC*, *MYB*, *MBS* and *G-box,* which shows its potential role in drought stress.

Using the online GSDS tool, the gene structure of all the *GmCAMTAs* was visualized in order to mutually compare their structural diversity. The length of GmCAMTA genes lie in the range between 3196 bp (*GmCAMTA14*) to 3947 bp (*GmCAMTA6*) with an average length of 3607 bp. The exons (yellow), introns (black lines) as well as 5’ and 3’ UTR (untranslated) regions (blue) of each gene are shown in figure 1B. A close observation of the number of exon-introns reveals a similar pattern, namely, three genes (*GmCAMTA7, GCAMTA10 and GmCAMTA15*) that are comprised of 12 exons; the rest have 13 exons and 12 introns of variable length, indicating their close mutual evolutionary relationship. This fixed numbering of intron and exon is a conserved characteristic of *CAMTA*, which is descended from ancestors and is also demonstrated in the *CAMTA* family of other species [23]. Similarly, except the last one/two exons, an ascending order in the exon size from 5’ to 3’ UTR can be observed across all the genes.

The four domains (CG-1, ANK (ankyrin repeat), IQ and CaMBD (CaM binding domain)) is the common conserved characteristic of all CAMTA TFs [23]. Scanning the amino acid sequences of GmCAMTAs with online protein domains illustrator tool showed the same four domains in all 15 members (figure 1C). GmCAMTA TFs through IQ (calcium-independent)/CaMBD (calcium-dependent) interact with Calmodulin, while through the CG-1 domain, they bind the DNA in a sequence-specific manner (*CGCG/CGTG*) at their promoter region. “ANK repeats” mediate protein-protein interactions. All these conserved domains, along with other properties, make GmCAMTA proteins, the “transcription factors”. The high sequence specificity is common in the Calmodulin binding domain of Arabidopsis and soybean CAMTAs as shown in figure 1D.

An ML (Maximum Likelihood) tree was constructed which traced the evolutionary relationship among the CAMTA families of soybean, Arabidopsis, maize and tomato (Figure 1E). Using MEGAX (molecular evolutionary genetic analysis), the evolutionary tree was constructed from the full length aligned CAMTA protein sequences of the four species. A total of 37 CAMTAs including six from Arabidopsis, 15 from soybean, seven from tomato and nine from maize clustered into four distinct groups, I, II, III and IV. GmCAMTA1–6, AtCAMTA1–3, SlCAMTA1 and 2 and ZmCAMTA3, 6, 7a and 7b might have co-evolved and thus clustered together in group I, representing the largest clade. Similarly, GmCAMTA10, 11, 14, 15, AtCAMTA4, SlCAMTA3, 4 and ZmCAMTA1 clustered in group III showing their mutually high homology. GmCAMTA8, 9, 12 and 13 grouped with AtCAMTA5, 6, SlCAMTA5, 6 and ZmCAMTA5 making clade IV. Two orthologs (GmCAMTA7 and SlCAMTA7) comprised group II, which is the smallest clade. Clustering in the phylogenetic reconstruction indicates more mutual similarity and probably weak homology to the members of the other three bigger clusters. It is noticeable that except II, in the rest of clades, CAMTAs of the same species are at the tips of the same branches and vice versa retaining their intraspecific homology.

### 2.3. miRNA Targets in GmCAMTA Transcripts

miRNA target prediction is important in finding their role in plant growth development in normal as well as stress conditions [18]. Keeping the Expectation score ≤5, the 639 miRNAs were scanned and miRNA, where the minimum E. value was selected using the online psRNATarget tool [39]. A total of 10 unique miRNAs were predicted which potentially target the *GmCAMTA* transcripts by inhibiting translation or through cleavage (Appendix A). Their length ranges between 19 bp (gma-miR1533) to 24 bp (gma-miR343b). The accessibility of target site (UPE), which is associated with identification of target site and energy required to cleave the transcript, varies from 11.8 (gma-miR1533) to 21.6 (gma-miR5780c). The translation of *GmCAMTA1* and *GmCAMTA2* transcripts is potentially inhibited by the common gma-miR5780c, while gma-miR6299 cleaves four *GmCAMTA8, GmCAMTA9*, *GmCAMTA12* and *GmCAMTA13*). Similarly, *GmCAMTA5* and *GmCAMTA 6* have potential targets for gma-miR1533 while *GmCAMTA11* and *GmCAMTA14* are predicted to be cleaved by gma-miR2111b. gma-miR9726 is predicted to cleave *GmCAMTA3* and gma-miR1522 *GmCAMTA15*. These in silico predictions require experimental validation, which would extend our understanding the mechanisms of *Ca-CaM-CAMTA*-mediated stress tolerance in plants.

### 2.4. Chromosomal Distribution and Regulatory Network of GmCAMTAs

The genome browser tool in NCBI (National Center for Biotechnology Information) mapped *GmCAMTA* genes to their respective chromosomes. The 15 *GmCAMTA* genes are unequally distributed over eight out of 20 chromosomes of soybean as shown in Figure 2. The figure depicts the complete size of each chromosome with the exact position of genes. Chromosome 8 has the highest number of *GmCAMTA* genes, i.e., four, while chromosome 7, 9, 11 and 18 have got only one in each. In prokaryotes, due to the absence of nucleus, transcription and translation occur simultaneously in coupling phase. On the other hand, translation of mRNA is always executed outside the nucleus in the eukaryotic cytoplasm and the proteins that work only in nucleus have a nuclear localization sequence/signal (NLS). Transcription factors also work in the nucleus; thus, after their translation in cytoplasm, they are directed to the nucleus which is mediated through their NLS. To find NLS, protein sequences of each GmCAMTA were submitted to the online cNLS mapper tool at http://nls-mapper.iab.keio.ac.jp/cgi-bin/NLS_Mapper_form.cgi. Keeping a cut off score of 5, at least one NLS in all GmCAMTAs and even more than one in some proteins were detected. Likewise, all of the six Arabidopsis CAMTAs possess only one NLS in the CG-1 domain of each protein [12]. In contrast, rice CAMTAs have one NLS in the C-terminal and another in N of CG-1 domain [40]. Experimental evidence shows that these domains perform diverse functions in the regulation of gene expression [41]. Appendix A shows the nuclear localization sequence in each of the 15 GmCAMTAs and predicts that all these transcription factors are localized in the nucleus.

In order to find the interaction network of GmCAMTA proteins to relate them with other pathways, the protein sequences were individually put in the STRING (Search tool for the retrieval of interacting gene/proteins) database, which predicted a number of interactors [42]. Thus, they can aid in linking proteins of interest to other pathways and could lead to the discovery of novel pathways as well. The STRING database displayed a network of ~10 interactors for each GmCAMTA protein among which some sets were redundant. Thus, a total of 48 unique proteins were predicted for 15 GmCAMTAs as shown in Figure 3 (Appendix A). This vast interaction network (experimentally determined and software predicted) indicates the complex regulatory upstream/downstream pathways of CAMTAs. However, these *in silico* predicted interactions require experimental validation. Besides these predicted interactions, their orthologues in other species such as Arabidopsis or other legumes could also be exploited to search other potential interactors in soybean.

### 2.5. GmCAMTAs as Early Drought Stress-Responsive TFs

The spatiotemporal expression in roots and leaves under 0, 1, 3, 6, 9 and 12 h of simulated drought stress is shown graphically in Figure 4. In roots, *GmCAMTA2* was highly expressed during 3 h of drought followed by *GmCAMTA7* and *GmCAMTA10*. In contrast, *GmCAMTA14* was downregulated during all the five time points followed by *GmCAMTA8, GmCAMTA9* and *GmCAMTA11*. Overall, these transcription factors upregulated abruptly during 1 and 3 h of stress and downregulated afterwards (Figure 4A). The expression profile of *GmCAMTAs* in leaves at different stress durations is different as compared to roots (Figure 4B). In leaves, they look uniform, except *GmCAMTA4,* which is the highest upregulated gene followed by *GmCAMTA5*, *GmCAMTA11* and *GmCAMTA12*. Interestingly, like roots, majority of these 15 genes retained the 3 h trend in the leaves as well. It is obvious that the expression was relatively high at 3 h, after which it decreased until 12 h.

The differential expression of *GmCAMTA* family is the result of their tightly regulated transcription. We speculate that in stress conditions, although the calcium ions continuously convey the stress signals through calcium signatures to the cytoplasm as well as the nucleus; however, the intensity/amount of these signals is weighed and adjusted by the next signal relaying molecules, such as CaM, before conveying to *CAMTA* transcription factors. In case, this signal transduction to *CAMTA* is continuing with the same intensity, yet after certain period (3 h in our case), *CAMTAs’* response is not the same throughout the course and seems to be unconcerned and even downregulated as the stress period continues. From the control samples (0 h) in leaves and roots, it is also obvious that the expression of all *CAMTAs* is active at all times. In brief, the spatiotemporal expression pattern revealed that *GmCAMTAs* are upregulated in the early phase of drought thus are early drought stress-responsive transcription factors.

### 2.6. Arabidopsis Overexpressing GmCAMTA12 Exhibited Enhanced Drought Tolerance

To evaluate the contribution of GmCAMTA12 protein in drought stress, we engineered Arabidopsis plants to constitutively express *GmCAMTA12* gene under 35 s promoter. Prior to Arabidopsis transformation, the *Agrobacterium tumefacians* strain EHA105 transformants were verified (through PCR) to harbor the overexpression cassette. (Appendix A). After floral dip, several overexpressing lines were obtained of which we selected two independent stable homozygous lines (OE5 and OE12) for functional analysis. The expression of *GmCAMTA12* in two transgenic Arabidopsis lines was validated through qPCR.

For drought assay, the two lines of OE *GmCAMTA12* (OE5 and OE12) and the wild type (WT) plants were subjected to drought stress by withholding water for two weeks and then re-watered as shown in Figure 5A. Initially, all the plants were growing normally until water was withheld. However, upon encountering drought, nearly all the wild type and transgenic Arabidopsis stopped growth, wilted and started turning yellow afterwards. After 14 days of continuous drought treatment, most of the wild type plants were completely dried as obvious from their phenotype (dried leaves). Unlike WT, most of the OE lines retained life processes, which was evident from chlorophyll they retained in their leaves. After re-watering, majority of transgenic Arabidopsis rejuvenated but most of the wild type did not. The plants were then allowed to grow under normal conditions until seed harvest. As expected, the seed yield of WT and transgenic lines was unequal and OE lines developed more seeds than wild type Arabidopsis. Under well-watered conditions, the survival rate in soil under drought was ~100%; however withholding water for two weeks and then re-watering, less than 60% WT plants survived while OE5 and OE12 showed about 83% and 87% survival rate as shown graphically in Figure 5D. Obviously, the constitutive overexpression of *GmCAMTA12* had enhanced the drought survival efficiency of transgenic Arabidopsis leading to better growth and development.

In their root length assay on ½ MS-mannitol medium as shown in Figure 5B, WT plants grew longer roots than OE plants at 0 mM concentration of mannitol; however, on mannitol, OE plants, specifically OE12, developed longer roots than WT at all three concentrations (50, 100 and 200 mM). Interestingly, the roots of OE12 plants were the longest at 200 mM mannitol. Root length (cm) is shown graphically in Figure 5E. As flaunted through root length assay, the comparatively longer roots of OE than WT were the result of *GmCAMTA12* overexpression in T3 seeds.

Between the two overexpression lines, OE12 performed better than OE5 in both drought assays, thus, we subsequently inoculated seeds of WT and OE12 on ½ MS with various concentrations of mannitol (50, 100, 150 and 200 mM) to evaluate the germination rate (Figure 5C). As expected, we observed higher germination rate of transgenic line OE12 than that of WT at all four concentrations of mannitol. The germination rate of OE12 was ~30% higher than WT at 50 mM mannitol. At 100 mM, the germination rate decreased in both types at a similar pace (~75% in OE and ~45% in WT). As the mannitol concentration increased to 150 and 200 mM, we saw a dramatic decline in the germination efficiency of WT (30% and 25%) as compared to OE (>65% and >60%) (Figure 5F). We can say that the constitutive expression on GmCAMTA12 has enhanced the germination efficiency of transgenic seeds under drought.

In order to check the performance of WT and OE lines at physiological level, the physiological indexes, such as proline and MDA contents, CAT activity and relative electrolyte leakage were determined in all plants subjected to stress. Under well-watered conditions, we observed no significant difference in the level of proline contents, which was quite low in WT and OE lines. In contrast, proline contents calculated in drought-stressed plants had significantly elevated. In WT, the average value of proline was ~400 µg/g, while in OE, it was recorded in the range of 850 to 900 µg/g (Figure 5G). Malonaldehyde (MDA) is a well-known biomarker for sorting out stress-induced membrane damage due to oxidative stress. MDA contents in WT and OE plants during normal conditions matched mutually but its level was doubled in the drought-stressed wt plants compared with drought treated OE lines (Figure 5H). Catalase (CAT) is a major antioxidant enzyme, which is accumulated during abiotic stresses and scavenges H_2_O_2_. The CAT activity in WT and OE plants under usual conditions was nearly equal (Figure 5I); however, drought treatment enhanced CAT activity in transgenic plants as compare to WT. During stress, electrolytes, specifically K^+^ ions leak out of the cells through various channels and thus damage due to stress could be monitored by comparing the electrolyte leakage in WT and transgenic lines. We determined relative electrolyte leakage (REL) in WT and OE lines during normal as well as drought conditions. REL % was nearly leveled under well-watered conditions, but was more pronounced in WT as compared to OE lines during drought (Figure 5J). Noticeably, the amino acid sequence analysis revealed high level of sequence similarity between GmCAMTA12 and AtCAMTA5. We can speculate that GmCAMTA12 being a transcription factor interacted with the downstream target genes (including AtCAMTA5 interactors) and modulated their expression in transgenic Arabidopsis, which contributed to their better performance under drought (determined at molecular level in later section).

### 2.7. GmCAMTA12 Overexpression Regenerated More Developed and Drought-Efficient Hairy Roots in Soybean

For further functional validation of *GmCAMTA12* in response to drought, the hairy roots system was exploited to overexpress the target gene in soybean. Prior to generating transgenic hairy roots, the *Agrobacterium rhizogenes* strain K599 cells harboring vector control (VC) and overexpressing *GmCAMTA12* construct (OE) were verified through gene-specific PCR (Appendix A). After 1–2 weeks of infection of soybean seedlings with K599 (VC and OE), hairy roots had started regenerating with various frequency. Prior to subsequent experiments, we made sure that the hairy roots were transgenic by using gene-specific PCR from the genomic DNA of a small piece of hairy roots. Using qPCR, we validated the overexpression of the target gene in OE roots which was over three times higher than the VC hairy roots. After hairy roots had become long and strong enough by growing for two more weeks, the primary roots were removed and the chimeric plants (with transgenic roots and non-transgenic shoots) were shifted to fresh vermiculite.

When the transgenic roots were ~10 cm long, the VC and OE chimeric soybean plants were extirpated from vermiculite and transferred to Hoagland solution as shown in Figure 6A. After 3 days of acclimation, both VC and OE chimeric plants were treated with 6% PEG6000. The plants started to wilt with leaves curling and shoot apexes drooping on encountering drought. However, the wilting was more in VC than in OE chimeric plants as VC leaves were more wilted. *GmCAMTA12* overexpression induced profuse hairy roots (Figure 6B) due to which the aerial non transgenic part of the OE chimeric soybean plant was also more developed than the VC chimeric plants. The roots were analyzed using the root scanning system (Figure 6C). The OE hairy roots showed higher values with total root length (Figure 6J), surface area (Figure 5K), root volume (Figure 6L), number of branches (Figure 6M) and projected area (Figure 6N).

The proline and MDA contents, CAT activity and relative electrolyte leakage were determined to check the impact of *GmCAMTA12* overexpression at physiological level. In control samples (Hoagland), proline contents had nearly equal amount in VC and OE hairy roots; however, under drought, proline content level was recorded to be significantly higher in OE hairy roots (Figure 6F). MDA shows the level of membrane damage as it is the final product of lipid peroxidation. In the absence of stress, MDA contents in OE type were slightly less than VC hairy roots; however, with PEG treatment, VC had substantial increase in MDA contents as compare to OE roots (Figure 6G). In contrast, CAT activity was significantly higher in OE hairy roots than VC under drought stress (Figure 6H). For REL, VC hairy roots had higher electrolyte leakage (%) than OE in response to drought (Figure 6I). Comparative physiology of OE and VC hairy roots shows that the overexpression of GmCAMTA improved the drought tolerance of OE by altering physiology. 

To analyze their growth efficiency under drought stress, 0.1 g of hairy root from VC and OE hairy roots was weighed under sterile conditions and inoculated on GM media containing four various mannitol concentrations (50, 100, 150 and 200 mM) including a control (0 mM mannitol) with three replicates of each type (Appendix A). All the plates were kept in dark in growth room with 28 °C for 10 days of culturing after which the fresh and dry weights were determined. The control samples of both types (VC and OE) hairy roots had nearly same fresh and dry weights; however, on stress media, OE hairy roots showed better performance as the weight of OE roots was more than VC roots at all four concentrations of mannitol (Figure 6D,E). It further flaunted that the overexpression of GmCAMTA12 has improved the drought survival efficiency of OE roots.

### 2.8. Expression Analysis of GmCAMTA12 Orthologues’ Regulatory Network in Arabidopsis

In Arabidopsis, *AtCAMTA5* is the orthologue of *GmCAMTA12;* thus, the regulatory network of *AtCAMTA5* was predicted with STRING database. To test our hypothesis, whether the overexpression of GmCAMTA12 TF in transgenic Arabidopsis would interact with genes in the regulatory network of *AtCAMTA5*, we analyzed the expression of 10 interactors predicted with STRING database (Figure 7A). Total RNA from drought treated WT and OE plants (Figure 7B) was isolated and reverse transcribed to cDNA. Using cDNA and gene-specific primers (Appendix A), we conducted a qPCR which deciphered the differential expression pattern of the 10 genes in wt and OE plants (Figure 7C). Among them, AtNIP30 (NEFA-interacting nuclear protein—*AT3G62140*), which in humans is involved in negative regulation of proteasomal protein catabolic process, is slightly upregulated in response to drought. AtWRKY14 (*AT1G30650*) encoding WRKY transcription factor 14 possesses a DNA binding domain and specifically interacts with *W box* (a common elicitor-responsive *cis*-element). WRKY14 is also nuclear localized transcription factor and regulates many important processes through gene regulation. *GmCAMTA12* overexpression upregulated *AtWRKY14,* which indicates that along with other transcription factors including WRKY TFs, the spectrum of CAMTA TFs regulatory networks is much wider than we think. Thus, the mutual interaction of CAMTA and WRKY should be further investigated. Peptdyl-prolyl *cis-trans* isomerase AtCYP59 (*AT1G53720*) mediates posttranslational modifications specifically protein folding. With *GmCAMTA12* overexpression, the upregulation of AtCYP59 is almost equal to that of WRKY14. AtANN5 (*AT1G68090*) is a calcium binding protein, plays role in pollen development and is induced by cold, heat, drought and salt stresses. As expected, its expression is relatively the highest among the 10 interactors. Serine racemase (AtSR—*AT4G11640*) is involved in serine biosynthetic pathway. Its expression also seems to elevate with *GmCAMTA12* overexpression in OE Arabidopsis. Elongator complex 3 (AtELO3—*AT5G50320*) is a part of Elongator multiprotein complex and regulates initiation and elongation of transcription. No apparent change in the expression level of AtELO3 in WT and OE was observed. Similar to ANN5, CaMHSP (Calmodulin Binding Heat Shock Protein—*AT3G49050*), also called Alpha/beta-Hydrolases superfamily protein, exhibited a higher transcript level. It is involved in the lipid metabolic pathway. B120 (G-type lectin S-receptor-like serine/threonine-protein kinase—*AT4G21390*) is involved in protein kinase activity and recognition of pollen. Its expression level was downregulated in transgenic Arabidopsis. *AT2G43110* (U3 containing 90S pre-ribosomal complex subunit) was also upregulated with the overexpression of *GmCAMTA12*. *AT3G19850* (BTB (for BR-C, ttk and bab) or POZ (Pox virus and Zinc finger)) domain-containing) mediates protein degradation by facilitating ubiquitination. Its expression level was elevated with *GmCAMTA12* upregulation. All of these genes possess a *CGCG/CGTG* motif in their promoter sequences, which also validates the sequence-specific binding of CAMTA TFs. Most of these interactions are based on text-mining and should be determined experimentally. 

### 2.9. GmCAMTA12 Overexpression Orchestrated Downstream Genes in Transgenic Hairy Roots

In order to find whether the constitutive overexpression of *GmCAMTA12* modulates the genes with which GmCAMTA12 TF interacts, we selected eight genes in the *GmCAMTA12* regulatory network in soybean predicted with STRING (Figure 7D). In chimeric soybean plants possessing VC and OE hairy roots (Figure 7E) treated with 6% PEG6000, we analyzed the expression pattern of the eight of the predicted interactors of *GmCAMTA12* to know their *GmCAMTA12*-mediated regulation. Using gene-specific primers (Appendix A), qPCR of the selected interacting proteins in transgenic hairy roots as well as non-transgenic leaves was carried out. The genes displayed a differential expression profile in the control and drought treated chimeric soybean plants. In VC roots (Figure 7H), GmNIP30 (NEFA interacting protein—*GLYMA19G40910*) was slightly upregulated in response to drought and its expression was indifferent to *GmCAMTA12* overexpression in OE roots (Figure 7I). GmPLA1-IId (Phospholipase A1-IId - *GLYMA12G15430*), involved in the lipid metabolic process, was upregulated in VC roots under drought stress, while in OE roots, its upregulation was two-fold higher than that in VC. *GmNAB* (Nucleic Acids Binding—*GLYMA18G40360*) was downregulated in VC roots while upregulated with *GmCAMTA12* overexpression in OE roots in response to drought. The expression of *GmELO* (Catalytic histone acetyltransferase subunit of the RNA polymerase II elongator complex—*GLYMA06G18150*) was the highest and equally expressed gene in both VC and OE roots. GmSR1 (Serine racemase-1 involved in D-Serine biosynthetic process—*GLYMA05G37930*) was upregulated in VC roots; however, with *GmCAMTA12* overexpression, GmSRI was downregulated in OE roots under drought. Similarly, GmSR2 (Serine racemase-2—*GLYMA08G01670*) was repressed in the absence of GmCAMTA12 overexpression; however, in OE roots under drought, it was slightly upregulated. Simultaneously, GmUC1 (uncharacterized, but possessing a *Myb*-DNA binding domain—*GLYMA20G32540*) transcript was slightly upregulated in VC but was significantly induced in OE hairy roots in response to drought. In contrast, GmUC2 (Uncharacterized2—*GLYMA17G07200*) was positively regulated in VC and negatively regulated in OE roots. Unlike the roots, the expression profile of these eight genes was nearly similar in chimeric soybean leaves possessing VC (Figure 7F) and OE (Figure 7G) hairy roots in response to drought. Five out of eight of these genes possess *CGCG/CGTTG* motif.

## 3. Discussion

Calcium as a ubiquitous secondary messenger orchestrates almost every cellular process in response to environmental stimulus. Plants employ the divalent cation, calcium (Ca^2+^) in relaying these endogenous (developmental) and exogenous (environmental) signals to appropriate cellular responses. Calcium alone specifically encodes a myriad of distinct signals by using spatial and temporal Ca^2+^ spikes as well as the frequency and amplitude of Ca^2+^ oscillations [10]. Next to the secondary messenger lie the signal relaying molecules including Calmodulin, which further tune the calcium signatures and pass them to *CAMTAs*. Calmodulin Binding Transcription Activators have a short history of two decades, which after first being reported have been genome-wide identified in numerous plant species [23]. *CAMTAs* are important in the sense that they are the transcription factors and an intermediate in the calcium-mediated stress signaling (*Ca-CaM-CAMTA*).

Earlier, the identification and expression analysis of Soybean CAMTAs was conducted [21]; however, their functional characterization remained unexplored. Thus, in order to better comprehend the role of *GmCAMTAs* in drought, and take a holistic snapshot, we first attempted to fill the research gaps through comprehensive in silico analysis of *GmCAMTA* family. Soybean possesses 15 *GmCAMTAs* [21], the second highest number after *Brassica napus*, which possesses 18 genes [24]. Interestingly, such a large stress responsive transcription factor family must have substantial contribution to the drought tolerance of soybean. *CAMTA* has important role in an array of biotic/abiotic stresses as reported in earlier studies in Arabidopsis [22,31,32,33,34,43], tomato [14,28], tobacco [18] and *Brassica napus* [24], Arabidopsis [36] and wheat [25]. Dissecting *GmCAMTAs* with various bioinformatics tools, their gene structure depicted 13 exon pattern, which is consistent with its orthologues in Arabidopsis, maize, tomato and others [23]. Promoter enrichment analysis revealed the *cis*-elements including *ABRE, DRE, G-box, W-box, WRKY, ARE* and *MYB*, all of which respond to various stresses [21]. *SlCAMTAs* are differentially expressed during the development and ripening of tomato and the presence of *ERE cis*-element in all *GmCAMTAs* provides the basis for *CAMTA* role in fruit development and ripening [28]. Light responsive elements (*G-box*) are common in all *GmCAMTAs* and we suggest that their role in light stress should be investigated [44]. Additionally, their specific stress-responsive *cis*-motifs should be tested individually or collectively in designing stress-inducible synthetic promoters.

Protein structure is important to know for understanding the action mechanism of protein. The major basic domains, CG-1, ANK, IQ and CaMB are common in all CAMTAs and a close look into the motif sequence of CaMB domain shows residues which are highly conserved across the species. Phylogeny of CAMTAs of four species traced the evolutionary relationship among the homologues as well as orthologues, which is consistent with the previous results [15,17,18,19,21,23,24]. Some homologues show more similarity and might have co-evolved (Figure 1). Later, the same set of proteins was found to interact with these homologues. A protein interaction network analysis found a few experimentally determined as well as predicted interactions together of 48 unique proteins of various pathways (Figure 3). Thus, we recommend to experimentally determine those interacting proteins to unveil GmCAMTA TFs’ link with the pathways of the interacting proteins. In Arabidopsis, 32 proteins were predicted with STRING in the interaction network of *AtCAMTAs* [23]. An experiment-based detailed map of all the *GmCAMTA* interactors would deepen our understanding of intricate mechanisms of *GmCAMTA*-mediated development and immunity against biotic/abiotic factors. Similarly, miRNAs are important transcriptional regulators by directly cleaving their target transcripts; thus, their potential target sites in *GmCAMTAs* were important to determine to understand *CAMTA*-mediated regulation. In silico analysis of *PtCAMTAs* revealed potential targets for four distinct miRNAs [18]. A set of 10 unique miRNAs was detected bringing more complexity in *CAMTA*-mediated stress response mechanisms in soybean (Appendix A). We recommend to experimentally determine the miRNAs targeting *GmCAMTAs*, which might also lead to the ability to engineer soybean and other crops against drought more accurately.

The expression analysis of *GmCAMTAs* in soybean leaf and root reveals that although all of them express constitutively, but *GmCAMTA2, GmCAMTA4, GmCAMTA5, GmCAMTA11* and *GmCAMTA12* are highly upregulated against drought. Secondly, almost all *GmCAMTAs* expressed and peaked in 3 h after which they were repressed indicating their early responsivity to drought stress (Figure 4). Based on the previous report [21], as well as our qPCR results, GmCAMTA12 was the common drought-efficient transcription factor, and thus, was selected to functionally characterize. As expected, overexpressing *GmCAMTA12* in Arabidopsis and hairy roots and drought assays thereof, validated the previous and current qPCR-based results. Interestingly, *GmCAMTA12* enhanced the drought survivability and growth performance of transgenic Arabidopsis (Figure 5), as well as hairy roots (Figure 6), which was validated at phenotypic, physiological and molecular level. Similar to G*mCAMTA12*, another transcription factor from soybean *GmNAC85* is also drought stress-responsive and its constitutive overexpression significantly enhanced the drought tolerance in transgenic Arabidopsis [45]. Similarly, RNAseq of soybean overexpressing *GmWRKY54* revealed that, *GmWRKY54* conferred drought tolerance by enhancing ABA(Abscisic acid)/Ca^2+^ signaling to close stomata as well as regulating numerous stress responsive transcription factors [6]. Similar results were recently reported about GmWRKY12 transcription factor [46]. In contrast, *AtCAMTA1*-mutant Arabidopsis exhibited enhanced drought sensitivity while also affecting the expression of other drought responsive genes [31]. A recent global transcriptome analysis using RNAseq revealed that a large number of genes involved in diverse stress responses are regulated either directly or indirectly by *AtCAMTA3* as about 3000 genes were misregulated in the *AtCAMTA3*-mutant Arabidopsis [34]. We recommend to experimentally verify the downstream targets of GmCAMTAs through protein-protein interaction experiments such as two-hybrid/pull down assays, which might link/lead to novel pathways. *GmCAMTA12* overexpression altered the expression of the genes in the regulatory network of *AtCAMTA5*, which is the orthologue of *GmCAMTA12* in Arabidopsis. All of the 10 interacting genes possess *CGCG/CGTG* motif in their 2000 bp upstream region validating the sequence-specific interaction of *GmCAMTAs* [12]. In conclusion, the better performance of OE Arabidopsis and chimeric soybean plants on MS-mannitol and Hoagland-PEG was validated at physiologic and molecular level. The comparison of Proline and MDA contents, CAT activity and relative electrolyte leakage between VC and OE hairy roots shown graphically, flaunts the improved development of chimeric plants with hairy roots overexpressing *GmCAMTA12*. This was further verified at a molecular level by determining the transcript level of the genes in the regulatory network of *GmCAMTA12*. The *GmCAMTA* overexpression should also be analyzed in other tissues including soybean flower and seeds as well as in various developmental stages to find its integrated role in processes besides stress tolerance. However, GmCAMTA12 nuclear localization, promoter-GUS analysis, CRISPR/cas9-mediated gene-knockout and global transcriptomics of stably transformed transgenic soybean and Arabidopsis are under investigation. Taken together, a more systematic approach should be adopted to decipher the integrated role of GmCAMTA TFs with the Calcium-Calmodulin signaling crosstalk in drought.

## 4. Materials and Methods

### 4.1. In Silico Analysis 

By keyword search (Gene ID/locus/accession no.) in three databases (NCBI https://www.ncbi.nlm.nih.gov/, Phytozome https://phytozome.jgi.doe.gov/pz/portal.html and Plantgrnnoble http://plantgrn.noble.org/), we retrieved the genomic, transcriptomic and proteomic sequences of the 15 members of *Glycine max CAMTA* gene family (Appendix A). Similarly, protein sequences of the corresponding orthologues in *Arabidopsis thaliana*, *Zea mays* and *Solanum lycopersicum* were also retrieved and a dataset was created for bioinformatics analyses (Appendix A).

The physico-chemical properties including molecular weights, theoretical isoelectric points, Aspartate+Glutamate, Arginine+Lysine, number of atoms, instability index, aliphatic index and GRAVY (Grand average of hydrophathicity) of all *GmCAMTA* proteins were calculated using the ProtParam tool in the ExPASy (https://web.expasy.org/protparam/).

To trace their evolutionary relationship, an ML tree was constructed with MEGAX using default parameters after multiply aligning the protein sequences of *Glycine max*, *Arabidopsis thaliana*, *Zea mays* and *Solanum lycopersicum*. 

The exons and introns along the length of *GmCAMTA* genomic sequences were monitored using the online tool GSDS (Gene Structure Display Server (http://gsds.cbi.pku.edu.cn/index.php) [13]. To check the *cis*-motifs within the regulatory regions of *GmCAMTAs*, (2kb upstream) 5’ UTR of each gene was resolved at online database PLACE (https://sogo.dna.affrc.go.jp/). The promoter sequences are in Appendix A. All the 15 *GmCAMTA* protein sequences were aligned with ClustalX and the conserved motifs were identified using the online SMART tool (Simple Modular Architecture Research Tool) (http://smart.embl-heidelberg.de/). The CaMBD in Arabidopsis and Soybean CAMTA family proteins was particularly analyzed for conserved sequences. The protein alignment is shown in Appendix A.

The potential miRNA targets along the length of *GmCAMTA* transcripts were predicted online at psRNATarget server (http://plantgrn.noble.org/psRNATarget/). To predict the proteins interacting with *GmCAMTAs*, each protein was separately submitted to STRING database (https://string-db.org/). The online prediction software displayed the proteins experimentally proved/hypothetical proteins interacting with *GmCAMTAs*.

The loci of *CAMTA* family over 20 chromosomes were determined using NCBI Genome Data Viewer at https://www.ncbi.nlm.nih.gov/genome/gdv/. Similarly, the subcellular localization of *GmCAMTA* transcription factors was determined by screening NLS in the protein sequence with online tool cNLS mapper at http://nls-mapper.iab.keio.ac.jp/cgi-bin/NLS_Mapper_form.cgi. NLS (nuclear localization signal/sequence) is an amino acid sequence, which, if present in a protein, indicates its nuclear localization. 

Primers for qPCR analysis were designed with Primer premier 5 software as well as NCBI online primer tool https://www.ncbi.nlm.nih.gov/tools/primer-blast/. The multiple alignments of *GmCAMTAs* CDS showed frequent conserved sequences; thus, to minimize ambiguity, the primers were designed with special care against the unique sites in each *GmCAMTA* CDS. Moreover, it was ensured that the primers amplify all alternative transcripts of respective gene. We attempted to design the pair targeting two exons amplifying a stretch of 200–300 bp cDNA. Those primers which failed to give amplification, created more than one peak or did not appear on agarose gel under UV were redesigned until corrected. Moreover, the primers specificity was determined with online nBLAST tool in NCBI https://blast.ncbi.nlm.nih.gov/Blast.cgi?PROGRAM=blastn&PAGE_TYPE=BlastSearch&LINK_LOC=blasthome. Suzhou GENEWIZ Biological Technology Services Company, China, synthesized all primers for this experiment listed in Appendix A.

### 4.2. Expression Analysis 

The seeds of soybean variety “Willliams 82” were washed with 75% ethanol and then sterile water. They were hydroponically cultured in Hoagland nutrient solution in a growth room with ~28 °C room temperature, 16–8 h (light-dark) photoperiod and a relative humidity of 50%. When the plants have opened their unifoliate leaf pair completely, they were subjected to 6% (PEG 6000) stress simulated drought. The plants were treated with drought for 1, 3, 6, 9 and 12 h in triplicate along with control (Hoagland). At the exact time points, we collected leaf and root samples, froze in liquid nitrogen and stored at –80 C for further processing.

Total RNA from the root and leaf samples (ground in liquid nitrogen) of soybean plants (treated with PEG for various time periods) was extracted using RNAiso Plus (Takara, Japan) according to the manufacturer’s protocol. RNA concentration was determined with NanoDrop2000Spectrophotometer (Thermo Fisher Scientific, Waltham, MA, USA). The quality of RNA samples was checked over 1% agarose gel using 0.5X TBE. (Tris/Borate/EDTA) buffer. Bright and clear bands of 25S/18S RNA showed the intact RNA. The samples that showed degraded RNA or relatively unclear bands were re-extracted until intact RNA was detected. The concentration of all good quality RNA samples was adjusted to ~500 ng/µL and it was reverse transcribed into cDNA using the PrimeScript RT reagent kit with gDNA Eraser (Takara, Japan) according to the company’s protocol. After the removal of genomic DNA with gDNA eraser, 1 µg total RNA was used as template to synthesize cDNA and stored at –20 °C. 

After the reverse transcription of RNA into cDNA and qPCR primer synthesis, qPCR was run according to the experimental design. Prior to the qPCR of all the samples, a standard curve was created with a reaction using a serially diluted cDNA in duplicate. Actin 11 was used as reference with leaf cDNA while Elongation factorα with that of root. Before determining the expression pattern of soybean *CAMTA* gene family in leaves and roots, the personal error was minimized by the standardization of experiment. An initial reaction was run on Stratagene Mx3000 P thermocycler and primers validity was determined first by confirming the single peak (amplification plot), the *C*_t_ value for each gene, and then by running the reaction product on agarose gel. Each product gave a single band parallel to right marker band. A 20 µL reaction mixture contained 10 µL SYBR Premix Ex Taq, 0.4 µL ROX Reference Dye II, 0.4 µL of each gene-specific Primer, 2 µL of cDNA and 6.8 µL ddH_2_O. High PCR efficiency is indispensable for robust and more precise RT-qPCR. The formula E = 10(−1/Slope)^−1^ was used to calculate amplification efficiency and the slope of the calibration curve. The primers having ~100% PCR amplification efficiencies were used. The RT-qPCR profile for our samples consisted of an initial denaturation of 95 °C for 2 min followed by 40 cycles of 94 °C for 5 s and 62 °C for 20 s. The fold change in relative expression level was evaluated using 2^−ΔΔ*C*t^ formula.

### 4.3. Gene Transformation and Drought Assays

From the bioinformatics as well as qPCR analyses, we selected the gene *GmCAMTA12* (*Glyma.17 G031900*) to clone and transform for functional validation. The complete CDS (2769 bp) of *GmCAMTA12* was PCR-amplified using cDNA as template, forward primer ACTAGTATGGCGAATAACTTAGCGG and reverse primer CCCGGGCTATGTCTTCAGTTGCATGTCAA. For amplification, LA Taq kit (Takara) was used according to the manufacturer’s protocol. The PCR profile was; an initial denaturation at 94 °C for 1 min followed 35 cycles of 94 °C for 30 s, 55 °C for 30 s and 72 °C for 150 s, and a final extension at 72 °C for 10 min. Using CT101 cloning kit (Transgene, Beijing, China), the gene was cloned into pEASY^®^-T1 cloning vector and transformed into Trans1-T1 chemically competent cells (Transgene) following heat shock method according to the manufacturer’s instructions. The positive clones were obtained on a selective LB-Kan^+^ agar plate and screened through vector and gene specific PCR as well as restriction. After double check, three separate clones were sequenced using M13 forward and reverse primers. When the sequences of all three clones were analyzed without any point mutation, the gene was restricted from pEASY^®^-T1 and sub-cloned by restricting from cloning vector using *SpeI* and *SmaI* (Takara) and ligated into the corresponding sites of expression vector pBASTA (with Kanamycin and Glyphosate resistance genes) using T4 Ligase (Promega, Madison, WI, USA) constituting the recombinant overexpression construct, pBASTA-*GmCAMTA12*. The pBASTA-*GmCAMTA12* construct was transformed into DH5α strain of *E.coli* (Transgene) and positive clones were obtained on a selective LB-Kan^+^ agar plate. After a double check (PCR and restriction) of single colonies harboring the overexpression construct, the colonies were preserved in sterile 80% glycerol. The recombinant vector was transformed into chemically competent *Agrobacterium tumefacians* (EHA105) and *Agrobacterium rhizogenes* (K599) by a 5 min liquid nitrogen treatment followed by a heat shock of 5 min at 37 °C. The positive clones of EHA105 were identified on YEP Kan+Rif plates while K599 were selected using YEP Kan+Strep plates. After PCR verification, the cells were preserved at –80 °C for future use. 

Arabidopsis Col seeds were kindly provided by Engineering Research Center of the Ministry of Education of Bioreactor and Pharmaceutical Development. To synchronize the germination of seeds, wt seeds were soaked in water and kept at 4 °C for 48 h and then sowed on a mixture of humus + vermiculite (3:7) in a growth room in dark at 23 °C. The germinated seedlings were then allowed to grow in 16 h photoperiod until 40 days with regular watering. We transformed the inflorescence (unopened flowers) of mature healthy Arabidopsis plants through Floral Dip method [47] and harvested T1 (Transgenic generation 1) seeds. The T1 seeds were germinated in excess under the same conditions as for wild type seeds. A primary screening of T1 plants (at six leaf stage) was carried out by spraying with 1% Basta (glyphosate) and the basta-survived plants were grown in separate pots in fresh soil (humus + vermiculite) under the same conditions as for wild type. Each plant represented a separate transformation event. The non-transformants died, while from the green plants, we extracted genomic DNA from leaves by 2 X CTAB and a PCR with gene specific primers using the gDNA as template was performed for secondary screening. T1 plants positive with PCR were grown to harvest T_2_ seeds. Similarly, T_3_ seeds from homozygous lines were harvested following the primary screening by 1% Basta and secondary with PCR to ensure stable transformation. 

The seeds of Arabidopsis were surface sterilized by keeping in 70% ethanol for 1 min followed by 50% bleach for 10 min after which the seeds were washed six times by sterile water. The wild-type and two T_3_ transgenic Arabidopsis lines were germinated under sterile conditions on half MS plates added with various mannitol concentration. For germination, we vernalized wt and OE seeds by keeping in dark for 3 days at 4 °C. Each petri plate was marked and divided in to three parts, one of which was allotted to wt while the other two for Line-5 (OE5) and Line-12 (OE12) seeds. The seeded plates were kept in a growth chamber with 16 h photoperiod and 23 °C temperature. The germination of each line was observed and recorded for one week and the germination rate was determined by the number of seeds germinated divided by the total number of seeds. For root length analysis, wt and T3 lines were first germinated on MS media under sterile conditions and one week old seedlings of same length were transferred to square plates containing MS with various concentrations of mannitol. The plates were placed in a rack vertically so that the roots grow downwards to ground. The root growth was observed and the plants were photographed. The seeds of wild-type and two T3 transgenic Arabidopsis lines were germinated and grown for one month in soil at 23 °C, watered regularly and then subjected to drought stress by withholding water for 14 days and photographed before, during and on drought recovery after re-watering.

Various physiological parameters like MDA [48] and proline contents [49] as well as CAT activity [50] and relative electrolyte leakage were determined. Cell membrane integrity is lost as K^+^ ions (a chief electrolyte) leaks out of the cell leading to cell death in stress. This measurement is an indicator of the cell damage caused by stress [51]. To determine REL, leaf sample (~2 g) was thoroughly rinsed with deionized water and then subjected to vibration for 2 h at 25 °C in a test tube containing 10 mL deionized H_2_O. Using a conductivity meter, we determined the conductivity of solution (C1). A second measurement of conductivity (C2) was taken after boiling the solution for 25 min and then cooled to RT. REL % was calculated using the formula (C1/C2) × 100.

Soybean hairy root is an established platform for the functional analysis of a gene by overexpression. For hairy roots, seeds of soybean variety JU 72 were washed with 75% ethanol and rinsed in sterile water a few times. The seeds were sown in pots containing autoclaved vermiculite and kept in a growth room with 16 h photoperiod and 28 °C temperature and watered regularly. In parallel, K599 from glycerol was streaked on YEP Kan + Strep plates and incubated at 28 °C for 48 h. A single colony was picked and inoculated in YEP containing Kanamycin and Streptomycin. The culture was incubated in a shaking incubator at 200 rpm and 28 °C for 48 h. A 100 µL of inoculum rom the culture was spread on YEP Kan + Strep plates and incubated at 28 °C for 48 h. When the seedlings had just sprouted, the cotyledons unfolded and the first unifoliate leaves had not yet appeared, they were ready to be infected with K599, harboring the overexpression construct. The *Agrobacterium rhizogenes* culture from the plate was picked with a needle and injected into the hypocotyl right at the base of cotyledon [52]. K599 harboring empty pBASTA were used to regeneration of VC (Vector Control) hairy roots. After infection, the seedlings were covered with a transparent plastic to ensure high humidity. Within two weeks, hairy roots started to sprout from the site of infection at variable frequencies with at least one root per seedling. The soybean plants with hairy roots were allowed to grow for two weeks after which excess of autoclaved vermiculite was added to cover the hairy roots and watered regularly. We extracted the genomic DNA from both type of the transgenic roots to confirm VC and OE through gene specific primers. The roots were scanned with a scanner and analyzed with the software.

After the hairy roots were ~10 cm long, the plants were uprooted, primary roots were cut and the chimeric plants (transgenic root and non-transgenic shoot) were transferred to fresh autoclaved vermiculite and watered regularly. It is important to notice that the shoot of each plant was cut after the second trifoliate leaves. Thus, the newly grown leaves were compared between the chimeric plants VC and OE transgenic roots. To check the performance of transgenic hairy roots under drought, plants with 10 cm long transgenic hairy roots were transferred to Hoagland solution and after acclimation, they were treated with 6% PEG6000. Proline and MDA contents, CAT activity and REL were determined in VC and OE hairy roots. Moreover, the 0.1 g of the hairy roots (VC and OE) were weighed in sterile conditions and grown on MS media at various concentrations of mannitol including 0, 50, 100, 150 and 200 mM. After 10 days, fresh and dry weights of the hairy roots were determined. The dry weight was measured keeping hairy roots at 60 °C overnight.

### 4.4. Expression Analysis of the GmCAMTA12 Orthologue’s Regulatory Network in wt and OE Arabidopsis 

The regulatory network of the orthologue of *GmCAMTA12* (*AtCAMTA5*) in Arabidopsis was predicted online with STRING database. The genomic and proteomic sequences of these 10 interactors are given in Appendix A. The primers for qPCR were designed against the 10 genes with Primer-BLAST tool (Appendix A). In parallel, RNA from wt and OE lines was extracted, quantified and reverse transcribed into cDNA with Takara kit following kit’s instructions. To determine the expression pattern of the 10 genes, qPCR was run using the primers of each gene with three biological replicates. *AtActin11* was used an internal reference. The data was analyzed with 2^-dd*C*t^ method.

### 4.5. Analysis of GmCAMTA12 Regulatory Network in Chimeric Soybean Plants

Using the STRING database, the interactors of *GmCAMTA12* in soybean were predicted. The primers for qPCR of eight genes designed with Primer-BLAST are given in Appendix A. The genomic and proteomic sequences of these 10 interactors are given in Appendix A. To determine the effect of overexpression of *GmCAMTA12* on its interacting proteins, RNA from all samples was extracted using RNAiso plus (Takara) and reverse transcribed into cDNA with Takara rt kit and qPCR was run for all samples in triplicate. Actin 11 was used as internal control. Relative expression level was determined using 2^-dd*C*t^ method.

## Figures and Tables

**Figure 1 ijms-20-04849-f001:**
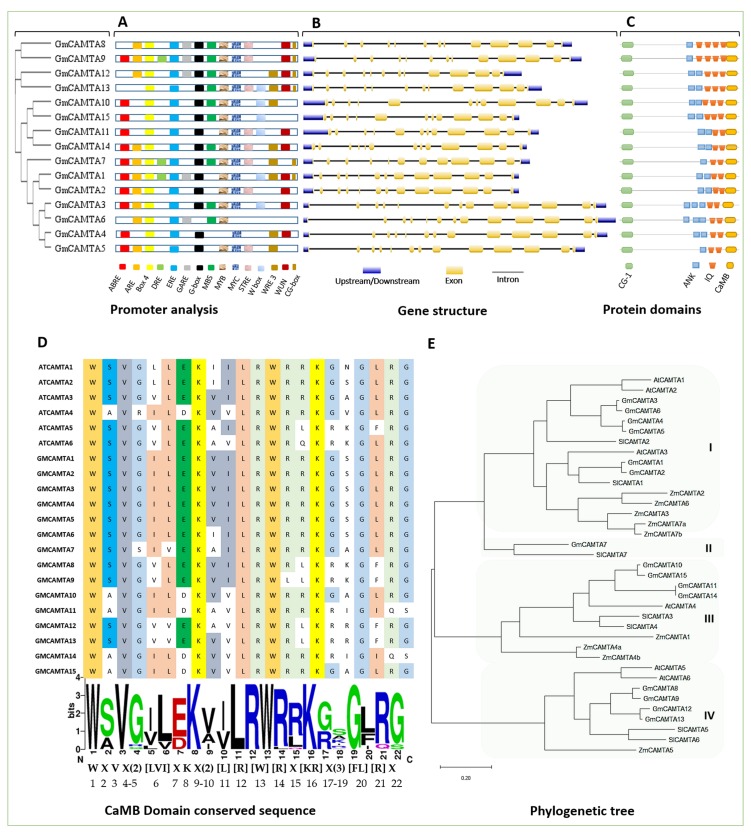
Comprehensive in silico analysis of the *GmCAMTA* family. (**A**) *Cis*-elements in the *GmCAMTA* promoter region. Each type of *cis*-motif present in *GmCAMTA* promoters is shown in unique color/color pattern. (**B**) Exon-intron assembly of *GmCAMTA* genes. (**C**) Domain organization of GmCAMTA proteins. Four different domains are represented in different colored shapes. (**D**) Alignment showing the conserved motif sequence of the Calmodulin Binding Domain of Arabidopsis and Soybean CAMTA TFs. Each conserved residue at the definite position along the row (throughout the orthologues) is shaded in unique color. The functional residues in CaMB domain of these CAMTAs are indicated in the motif below the alignment. In the square brackets “[ ]” are the amino acids allowed in this position of the motif; “X” represents any amino acid and the round brackets “( )” indicate the number of amino acids. (**E**) ML (maximum likelihood) phylogenetic tree.

**Figure 2 ijms-20-04849-f002:**
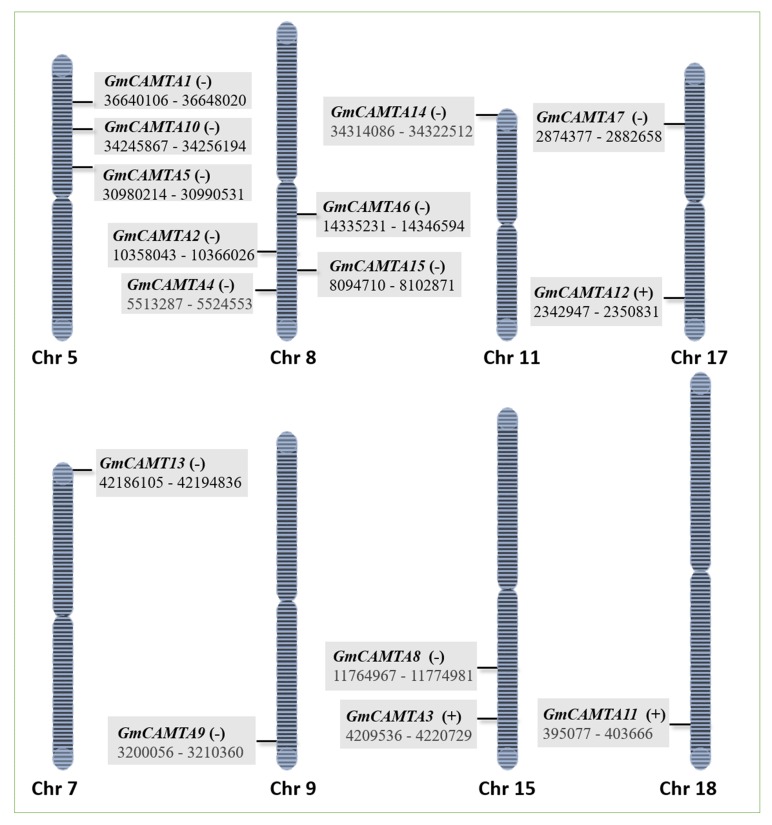
Chromosomal distribution of *GmCAMTA* genes. The 15 *GmCAMTA* genes are located on chromosome 5, 7, 8, 9, 15, 17 and 18.

**Figure 3 ijms-20-04849-f003:**
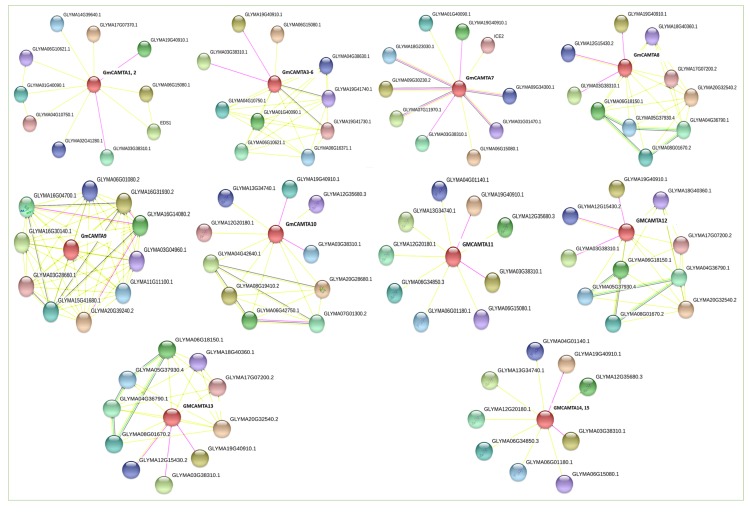
Protein-protein interaction network of GmCAMTAs detected in silico using the STRING database (https://string-db.org/). The experimentally determined interactions are denoted by pink strings, interactions on the basis of textmining are indicated by yellow strings while interactions on the basis of gene neighborhood shown by green strings.

**Figure 4 ijms-20-04849-f004:**
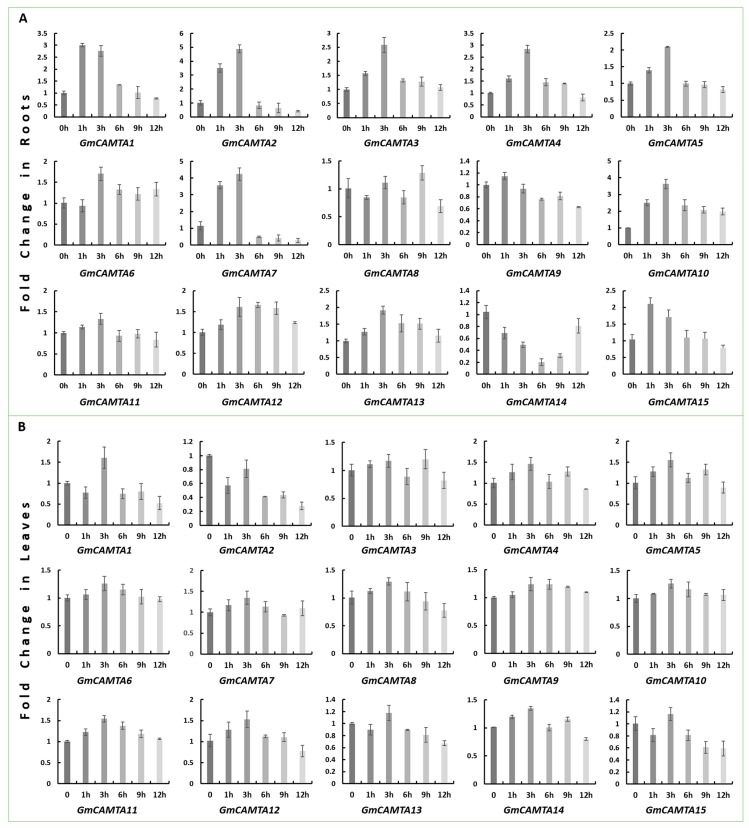
Spatiotemporal expression analysis of GmCAMTA in drought. (**A**) Relative fold expression of *GmCAMTAs* in Roots. Soybean plants were treated with 6% PEG6000 in Hoagland’s solution for five different durations (1, 3, 6, 9 and 12 h) along with a control (0 h). (**B**) Relative fold expression of *GmCAMTAs* in Leaves.

**Figure 5 ijms-20-04849-f005:**
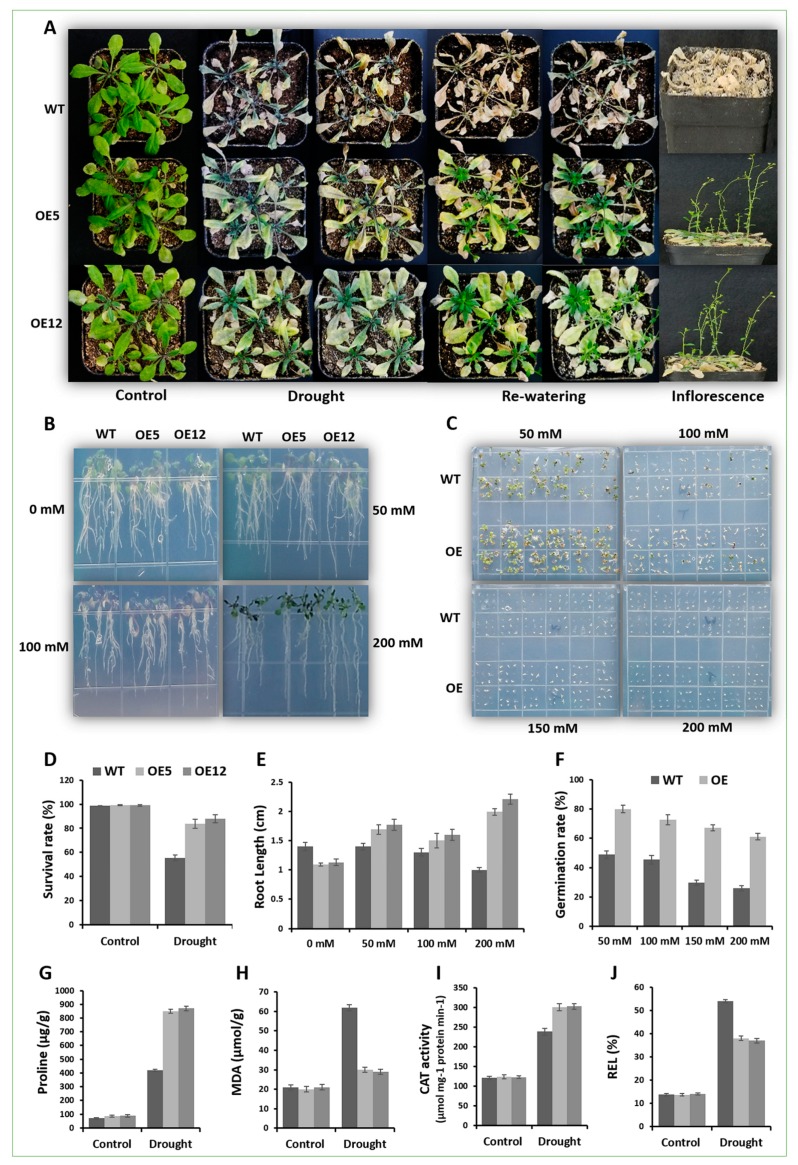
Phenotypic and physiological assays of wild type (WT) and OE under drought stress. (**A**) Drought assay of wild type (WT) and transgenic (OE5 and OE12) Arabidopsis grown on soilrite subjected to 14 days of drought stress. (**B**) Root length assay on MS-mannitol. (**C**) Germination assay on MS-mannitol. (**D**) Column chart showing the survival rate (%) of WT and OE in soil. (**E**) Root length (cm) of WT, OE5 and OE12 lines. (**F**) Germination rate (%) of WT and OE plants. (**G**) Proline contents, (**H**) malonaldehyde (MDA) contents, (**I**) CAT activity and (**J**) relative electrolyte leakage (REL %) of WT and OE plants in control and drought treatment.

**Figure 6 ijms-20-04849-f006:**
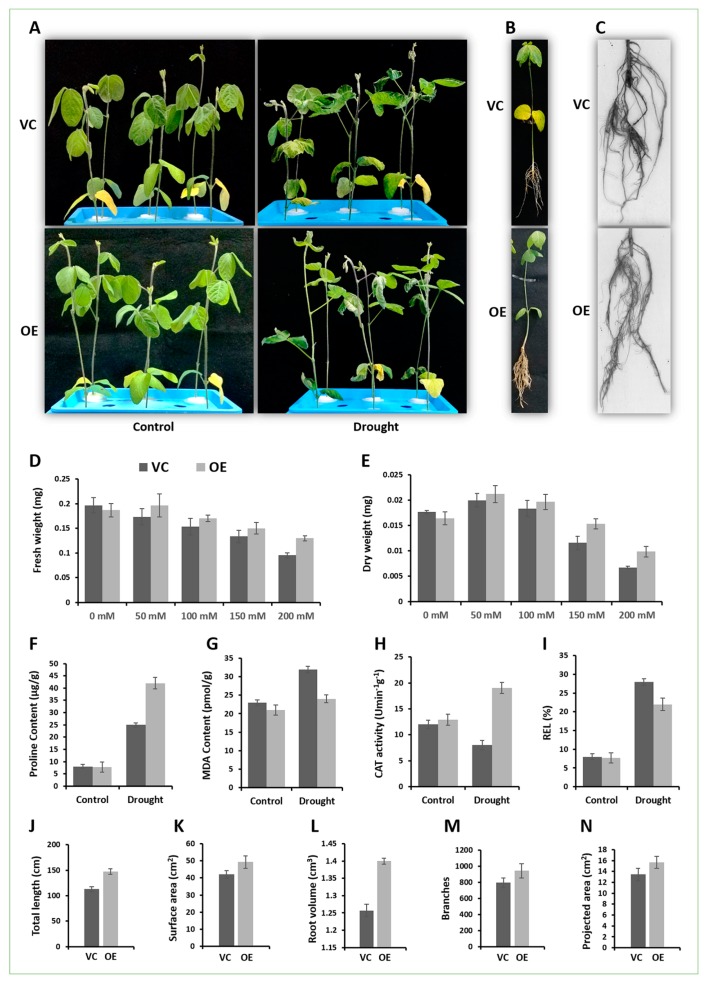
Comparative phenotype and physiology of chimeric sobean plants. (**A**) Chimeric soybean plants bearing VC (Vector control) roots and OE (overexpressing GmCAMTA12) roots in control and drought conditions. (**B**) Comparison of chimeric soybean plants having VC and OE roots. (**C**) VC and OE hairy roots observed with a root scanner. (**D**) Fresh and (**E**) dry weight of VC and OE hairy roots after culturing on MS-mannitol for 10 days. Comparison of VC and OE hairy roots in terms of (**F**) Proline contents, (**G**) MDA contents, (**H**) CAT activity and (**I**) relative electrolyte leakage between VC and OE hairy roots. (**J**) Analysis of the total length, (**K**) surface area, (**L**) root volume, (**M**) number of branches and (**N**) projected area.

**Figure 7 ijms-20-04849-f007:**
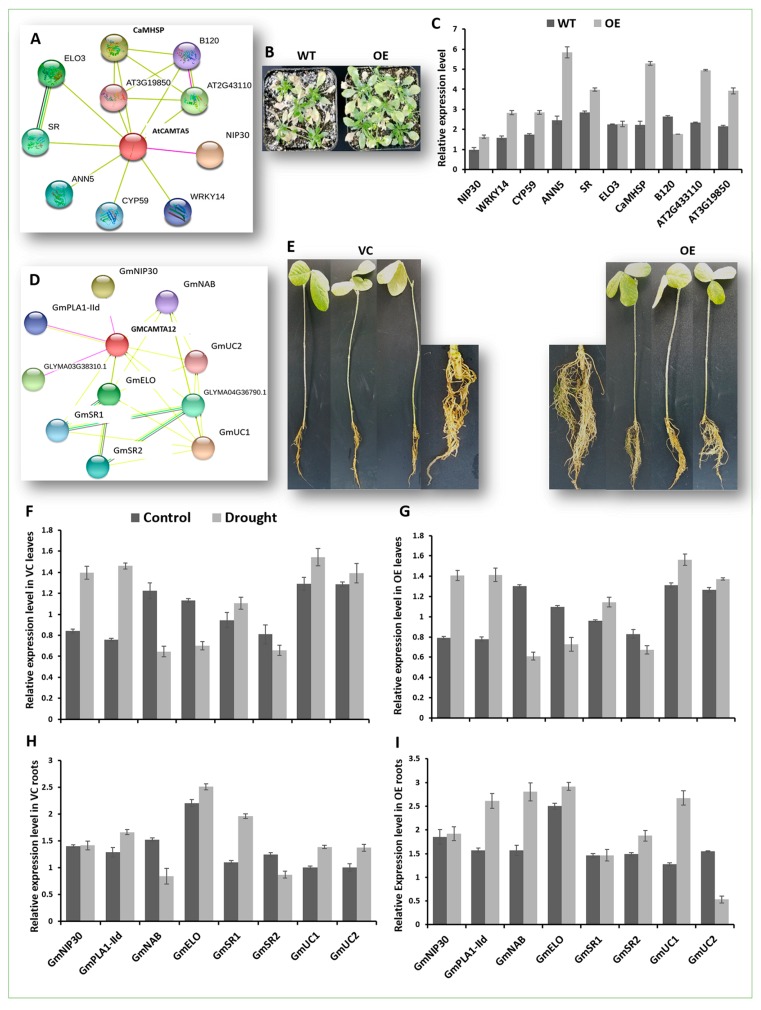
Expression analysis of genes in the regulatory network of *GmCAMTA12* and *AtCAMTA5*. (**A**) STRING-predicted regulatory network of *AtCAMTA5*. (**B**) Drought treated WT and OE Arabidopsis. (**C**) Expression profile of the 10 interactors of *AtCAMTA5* in WT and OE Arabidopsis. (**D**) STRING-predicted regulatory network of GmCAMTA12 in soybean. (**E**) Chimeric soybean plants with VC and OE hairy roots. (**F**) Expression analysis of the eight interactors of *GmCAMTA12* in VC and (**G**) OE leaves. (**H**) Expression analysis of the eight interactors in VC and (**I**) OE roots.

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
