# Peer review of "Overexpression of GmCAMTA12 Enhanced Drought Tolerance in Arabidopsis and Soybean"

_ijms, 2019, doi:10.3390/ijms20194849_

Round 1

Reviewer 1 Report

The manuscript by Noman et al. entitled ‘Overexpression of GmCAMTA12 enhanced drought

tolerance in Arabidopsis and Soybean’ is an updated version of previously submitted ‘In silico Investigation and Spatiotemporal Expression of Soybean CAMTAs Provide Insights as Early and Rapid Drought Stress-Responsive Transcription  Factors’ reports on bioinformatic analysis and gene expression of soybean CAMTA TFs’. In the resubmitted manuscript, the authors added new set of data on the overexpression of one of the CAMTA TFs in Arabidopsis and hairy roots of soybean. Even though the overexpression data may support authors conclusion on the role of CAMTA TFs under drought stress, this manuscript still cannot meet the standard of International Journal of Molecular Sciences in its current form. It requires thorough revision including language editing and improving data/figure formatting.

More comments

1.      The quality of all the figures and pictures is very poor. They very blurred. The numbers and letters are not readable. The resolution of plant pictures is very poor. The authors have stated in their response to previous review that the ‘Figure redrawn with improved quality’, but this is not the case. Authors may consider using a special software used for making graphs for example, sigma plot, Adobe Illustrator etc.

2.      The way the figures are labeled, and the figure legends need to be simplified. It is not clear what some figure panels represent because they are explained in the legend. For example, Figure 4-l, 4-m and 4-n are not explained in the legend.

3.      Figure 5 panels 5-s and Figure 5-t are not visible at all.

4.      The language needs to be thoroughly edited by a professional or native English speaker.  The authors have stated in their response to previous review that the ‘Language has been improved’, but this is not the case. This is required to make the manuscript understandable by all readers.

5.      Looking at the effect of drought on CAMTA TFs expression, CAMTA1,2,3,4,5,7, 10 appear more highly induced by drought than CAMTA12. Why did the authors picked CAMTA12 for the overexpression?

6.       In Figure 4-e, the plants look permanently wilted, but in 4-f, there is inflorescence  from recovered plants. It is questionable if the inflorescence in 4-f are from plants in 4-e?

7.      Table 1 and Table 2 need to be included as supplementary information, not in the main manuscript

8.       Figure 2 a, chromosomal distribution needs to be included as supplementary information

9.      Abbreviation may not be necessary. It can be spelt out when it first appears.

Author Response

Response to Reviewer 1

In the resubmitted manuscript, the authors added new set of data on the overexpression of one of the CAMTA TFs in Arabidopsis and hairy roots of soybean. Even though the overexpression data may support authors conclusion on the role of CAMTA TFs under drought stress, this manuscript still cannot meet the standard of International Journal of Molecular Sciences in its current form. It requires thorough revision including language editing and improving data/figure formatting.

Dear Sir, we are grateful to you for taking your time to thoroughly check our resubmitted manuscript. Your valuable comments have helped a lot to improve the manuscript. We have addressed and accepted all your comments and the manuscript has been modified according to your kind suggestions. The language has been thoroughly revised by a native speaker, improved quality figures have been included and data/figure formatted according to your kind suggestions. The modifications can be tracked in the revised MS.

More comments

The quality of all the figures and pictures is very poor. They very blurred. The numbers and letters are not readable. The resolution of plant pictures is very poor. The authors have stated in their response to previous review that the ‘Figure redrawn with improved quality’, but this is not the case. Authors may consider using a special software used for making graphs for example, sigma plot, Adobe Illustrator etc.

Dear Sir, we are sorry for low quality figures and pictures in the previous version. Clear figures with improved quality have been added in revised version.

The way the figures are labeled, and the figure legends need to be simplified. It is not clear what some figure panels represent because they are explained in the legend. For example, Figure 4-l, 4-m and 4-n are not explained in the legend.

Thanks for guidance. Figures labeling and legends have been simplified and each part has been explained in the legend.

Figure 5 panels 5-s and Figure 5-t are not visible at all.

Figure 5 has been included with improved quality.

The language needs to be thoroughly edited by a professional or native English speaker. The authors have stated in their response to previous review that the ‘Language has been improved’, but this is not the case. This is required to make the manuscript understandable by all readers.

Language has been thoroughly checked and edited by a native speaker. All the corrections could be tracked.

Looking at the effect of drought on CAMTA TFs expression, CAMTA1, 2, 3, 4, 5, 7, 10 appear more highly induced by drought than CAMTA12. Why did the authors pick CAMTA12 for the overexpression?

Based on our qPCR results, GmCAMTA4, GmCAMTA5, GmCAMTA11 and GmCAMTA12 were more efficient in response to PEG. However, Wang et al. (2015) reported GmCAMTA12 and GmCAMTA14 to be more highly induced against the same stress. Considering both results, we picked the common transcription factor (GmCAMTA12) for overexpression analysis.

In Figure 4-e, the plants look permanently wilted, but in 4-f, there is inflorescence from recovered plants. It is questionable if the inflorescence in 4-f are from plants in 4-e?

The inflorescence in Figure 4-f were not from the plants in 4-e. The WT plants in 4-e were permanently wilted and did not recover from drought. Thus, 4-f was showing drought recovered WT plants of another pot. However, in the revised MS, we have replaced it with the pot (previously 4-e) showing the permanently wilted plants which didn’t recover after re-watering.

Table 1 and Table 2 need to be included as supplementary information, not in the main manuscript

Thanks for correction. Both the tables have been excluded from the main manuscript and included as supplementary information.

Figure 2 a, chromosomal distribution needs to be included as supplementary information

Thank you, Sir. Figure 1a (chromosomal distribution) has been included as supplementary information.

Abbreviation may not be necessary. It can be spelt out when it first appears.

Alright Sir. All abbreviations have been spelt at their first use, so abbreviation table has been removed.

Reviewer 2 Report

I'm not sure why the paper "Overexpression of GmCAMTA12 enhanced drought tolerance in Arabidopsis and Soybean" got results in Arabidopsis and soybean. I inform you that this paper cannot be printed with two important issues.

The data in Table 1, Figure 1, Table 2, and Figure 2 are data collected rather than the results of the study. The results of the study with two plants are not considered to be suitable for the purpose of this paper.

In both plants, GmCAMTA12 was fixed in the F3 line, which is called the homogeneous system. However, it is difficult to say that it is homogeneous to the F3 line because only PCR results are presented in the additional data. Therefore, the result of the study of homogenized line is further required. Finally, the following results are considered ironic.

Author Response

I'm not sure why the paper "Overexpression of GmCAMTA12 enhanced drought tolerance in Arabidopsis and Soybean" got results in Arabidopsis and soybean. I inform you that this paper cannot be printed with two important issues.

Dear Sir, Thanks a lot for checking the quality of work in our manuscript.  Below are our answers to your valuable comments.

The data in Table 1, Figure 1, Table 2, and Figure 2 are data collected rather than the results of the study. The results of the study with two plants are not considered to be suitable for the purpose of this paper.

Table 1, Figure 1 and Table 2 and Figure 2 display data based on in silico analyses of GmCAMTA family. The reason we included these data in manuscript is to provide the sequential features of soybean transcription factors linked with the expression and overexpression analyses. As this part of project consisted of three main steps, i.e., comprehensive in silico analyses using bioinformatics software, expression analysis using qPCR and overexpression analyses using Arabidopsis and soybean hairy roots systems. However, in the revised version, we have shifted table 1 and table 2 and figure 2a from the main manuscript to supplementary information.

As far as, studying the target gene by overexpression in the two plants is concerned, we chose Arabidopsis for pre-testing the target gene because Arabidopsis is a well-studied small dicotyledonous model plant and has been exploited (for manipulation) over last 20 years in molecular studies. Its short life cycle, small genome, self-progeny and established convenient transformation protocols allow the quick analysis of a transgene. Arabidopsis offers the ability to test hypothesis quickly and efficiently.

Secondly, soybean was used to study the overexpression analysis, because, the origin of the gene is soybean. We isolated GmCAMTA2 CDS from soybean. As the project aims at improving the drought tolerance of soybean, and this transcription factor family has been extensively studied in Arabidopsis as well as reported in other plants including soybean. In soybean, CAMTA family was proposed to be involved in drought thus an extension to that study, we overexpressed GmCAMTA12 in soybean hairy roots to validate its drought-efficiency.

Thus, we overexpressed GmCAMTA12 in both plants to compare and double check.

In both plants, GmCAMTA12 was fixed in the F3 line, which is called the homogeneous system. However, it is difficult to say that it is homogeneous to the F3 line because only PCR results are presented in the additional data. Therefore, the result of the study of homogenized line is further required. Finally, the following results are considered ironic.

GmCAMTA12 was fixed only in Arabidopsis F3 line and not in soybean plant. Among several transgenic lines, two of the overexpression lines (OE5 and OE12), whose expression was analyzed using qPCR, were proceeded to further experimentation for drought assays. The purpose of presenting PCR results in additional data was just to show the band size of GmCAMTA12 full length CDS.

In contrast, GmCAMTA12 was not stably transformed in soybean but was overexpressed only in hairy roots. Those plants were termed as chimeric soybean due to transgenic roots and non-transgenic shoots. Thus, there was no F2 or F3 line in the case of soybean. Prior to drought assays, a small portion of hairy roots (genomic DNA) was verified to be transgenic through PCR.

Round 2

Reviewer 2 Report

I am recognizing that this paper has been revised so much that it is appropriate for the Journal. However, the author does not have a clear purpose. Because comparing to soybean and Arabidopsis is similar to the relationship between basic science and applied science. In this paper, we have transformed soybean and Arabidopsis to "GmCAMTA12" and display the result of overexpression in T3. If so, fig1 and fig2 use the genebank results, so I would recommend going to the supplement. And most importantly, the research compared to soybean and Arabidopsis is wrong. Therefore, if the results are separated and summarized, it is judged that more advanced results can be printed in this Journal.

Author Response

Dear Sir, thanks a lot for your kind suggestions.

The structure and topical sequence of the manuscript were actually designed according to the project. The project started by the in silico analyses followed by expression analysis and then overexpression analysis of one of the drought-efficient transcription factor. Thus fig.1 and fig.2 were added as comprehensive bioinformatics analysis of GmCAMTA family that was recognized as research gap. Secondly, thgene was overexpressed in Arabidopsis in order to have a quick analysis in model plant prior to screening in soybean. Lastly, the results presented are not only a mere comparison of Arabidopsis and soybean but we wanted to sort out the downstream genes of GmCAMTA12 (in soybean) and of its orthologue in Arabidopsis.